# A giant NLR gene confers broad-spectrum resistance to *Phytophthora sojae* in soybean

Weidong Wang [1,6], Liyang Chen[1,6], Kevin Fengler[2], Joy Bolar[2], Victor Llaca [2], Xutong Wang[1], Chancelor B. Clark[1], Tomara J. Fleury[3,4], Jon Myrvold[2], David Oneal[2], Maria Magdalena van Dyk[2], Ashley Hudson[2], Jesse Munkvold[2], Andy Baumgarten[2], Jeff Thompson[2], Guohong Cai[3,4], Oswald Crasta[2,5], Rajat Aggarwal [2✉] & Jianxin Ma [1✉]

*Phytophthora* root and stem rot caused by *P. sojae* is a destructive soybean soil-borne disease found worldwide. Discovery of genes conferring broad-spectrum resistance to the pathogen is a need to prevent the outbreak of the disease. Here, we show that soybean *Rps11* is a 27.7-kb nucleotide-binding site-leucine-rich repeat (NBS-LRR or NLR) gene conferring broad-spectrum resistance to the pathogen. *Rps11* is located in a genomic region harboring a cluster of large NLR genes of a single origin in soybean, and is derived from rounds of unequal recombination. Such events result in promoter fusion and LRR expansion that may contribute to the broad resistance spectrum. The NLR gene cluster exhibits drastic structural diversification among phylogenetically representative varieties, including gene copy number variation ranging from five to 23 copies, and absence of allelic copies of *Rps11* in any of the non-*Rps11*-donor varieties examined, exemplifying innovative evolution of NLR genes and NLR gene clusters.

[1] Department of Agronomy, Purdue University, West Lafayette, IN 47907, USA. [2] Research and Development, Corteva Agriscience™, Johnston, IA 50131, USA. [3] Department of Botany and Plant Pathology, Purdue University, West Lafayette, IN 47907, USA. [4] Crop Production and Pest Control Research Unit, USDA, ARS, West Lafayette, IN 47907, USA. [5] R&D, Equinom, Inc., Indianapolis, IN 46268, USA. [6]These authors contributed equally Weidong Wang, Liyang Chen. ✉email: rajat.aggarwal@corteva.com; maj@purdue.edu

*P*hytophthora is a genus of plant-damaging oomycetes with more than 170 identified species[1], many of which are plant pathogens of considerable economic importance such as *Phytophthora infestans* that caused the Irish potato famine and subsequent Irish diaspora[2], and *Phytophthora sojae* that is responsible for annual yield losses estimated at ~$200 million in the United States and 1–2 billion worldwide[3,4]. In general, plant diseases caused by *Phytophthora* are difficult to control chemically[5], thus deployment of resistant varieties is an effective and environmentally friendly strategy to manage *Phytophthora* diseases[6].

In recent decades, a few dozen resistance-to-*P. sojae* (*Rps*) loci/alleles have been mapped to a number of soybean genomic regions enriched with nucleotide-binding site leucine-rich repeat (NBS-LRR or NLR) genes according to the soybean reference genome assembly v2.0 (soybease.org). Nevertheless, many of those loci/alleles have become ineffective to newly evolved isolates of the pathogen[7] and few *Rps* loci/alleles have been isolated and/or functionally validated. Therefore, effective introgression/pyramiding of specific *Rps* genes into elite soybean cultivars for sustained resistance remains challenging.

Here we show that soybean *Rps11* confers broad-spectrum resistance to *P. sojae*. A 27.7 kb NLR gene is pinpointed as the candidate gene of *Rps11* by gap-free sequence-based fine mapping and expression analyses, and its function is validated by stable transformation. We find that the exceptionally large size of *Rps11* results from rounds of inter- and intra-specific unequal recombination. Comparative genomic analysis reveals drastic structural and gene copy number variation among diverse soybean varieties. The isolation of *Rps11* will accelerate the deployment of the gene for soybean protection.

## Results

### The *Rps11* locus alone is responsible for the broad-spectrum resistance.

We have recently identified an *Rps* locus, designated *Rps11*, in a soybean landrace, PI 594527, which carries resistance to 12 *P. sojae* races examined[8] (Supplementary Fig. 1). Based on the responses of 209 $F_{2:3}$ families derived from a cross between PI 594527 and a susceptible variety Williams to *P. sojae* race 1, *Rps11* was initially mapped to a 348 kb genomic region on chromosome 7[8], with 12 sequencing gaps of unknown sizes according to the Williams 82 reference genome assembly v2.0. Nevertheless, the initial mapping only indicates that *Rps11* confers resistance to race 1.

To evaluate the *Rps11* resistance, we inoculated a subset of the $F_{2:3}$ families, including 14 families of the *Rps11/Rps11* (PI 594527) genotype and 14 families of the *rps11/rps11* (Williams) genotype with 14 *P. sojae* races separately, and observed perfect genotype–phenotype associations (Supplementary Table 1), suggesting that the *Rps11* locus alone is responsible for the broad spectrum of resistance carried by PI 594527. We then evaluated $F_5$ recombinant inbred lines (RILs) derived from several $F_2$ homozygotes (*Rps11/Rps11*) with 158 additional *P. sojae* isolates collected from soybean fields across Indiana and found that those RILs carry resistance to 127 (80%) isolates (Supplementary Data 1). These isolates are widespread across the state (Supplementary Fig. 2), but their race types remain to be characterized.

### Gap-free sequence-based fine mapping of *Rps11* with a sizable population.

As the *rps11* region in Williams 82, which is identical to the corresponding region in Williams, had not been fully assembled, and also because this region is enriched with NLR genes and thus thought to be highly labile, it was difficult to design markers for fine mapping. In order to clone *Rps11*, we sequenced and assembled the PI 594527 genome, by a combination of PacBio sequencing (PacBio Biosciences, Menlo Park, CA, USA), Bionano optical maps (Bionano Genomics, San Diego, CA, USA), and 10× Genomics chromium

genome linked-read sequencing (10× Genomics, Pleasanton, CA, USA), into 43 scaffolds with a scaffold N50 of 26.4 Mb (see details in "Methods"). In the course of PI 594527 sequencing, Chu et al.[9] developed a high-quality Williams 82 genome assembly (dubbed v3.0), which reveals a number of assembly errors in the *rps11* region in the Williams 82 assembly v2.0 (Supplementary Fig. 3).

To eliminate potential inaccuracy in the initial *Rps11* mapping result caused by possible assembly errors, we annotated and compared the *Rps11* region harboring the entire NLR clusters from the PI 594527 genome assembly and the corresponding region from the Williams 82 genome assembly v3.0. The ~648 kb *Rps11* region from PI 594527 is composed of 12 NLR genes (dubbed R1–R12) (Fig. 1a). Of these, R1, R4, R6, R7, R9, R11, and R12 are intact and are predicted to encode large NLR proteins containing 2315–2463 amino acids; R2, R3, R5, and R8 are truncated, with 2 to 4 exons lost at their 3′-ends; and R10 carries a 1.4 kb insertion in its first exon (Supplementary Fig. 4a). Only R1, R4, R6, R9, and R12 are expressed in stems as revealed by RNA sequencing (RNA-seq) (Supplementary Fig. 4b). The ~522 kb corresponding region from the Williams 82 assembly v.3.0 is composed of eight NLR genes (dubbed r1–r8). Among these genes, R8-r7 appear to be the only allelic pair between the two genomes (Fig. 1a). In fact, only a small proportion of the region is shared between PI 594527 and Williams 82 as syntenic blocks, which nevertheless allowed us to design markers for fine mapping (Fig. 1a).

Using two boundary markers, SSR286 and SSR320, in the conserved regions adjacent to the NLR clusters in PI 594527 and Williams 82, we genotyped 17,050 $F_4$ plants derived from heterozygous (*Rps11/rps11*) $F_3$ plants of the initial mapping population and identified 43 recombinants between the two markers (Supplementary Fig. 5). These recombinants were then genotyped with eight additional markers distributed in the syntenic blocks between the boundary markers and inoculated with *P. sojae* race 1. Combining the genotypic and phenotypic data from these 43 recombinants eventually defined the *Rps11* locus to a 151 kb region of PI 594527, harboring four NLR genes (R5, R6, R7, and R8) (Fig. 1b and Supplementary Fig. 5). As most of the fine-mapped region is not shared by the two parental lines (Fig. 1a), it is apparent that finer mapping would be inefficient or ineffective in pinpointing *Rps11*. Thus, we decided to explore an alternative approach to identify the candidate gene for *Rps11*.

### The candidate gene for *Rps11* is pinpointed through monitoring gene expression.

Given the fact that functional NLRs must be translated into proteins, as intracellular receptors, to mediate the specific recognition of pathogen avirulence effectors and activate immune responses[10,11], we monitored the expression of R5, R6, R7, and R8 in the stems inoculated with *P. sojae* race 1 and the stems uninoculated in PI 594527. Of the four genes, only R6 was expressed in both inoculated stems and uninoculated stems, whereas R5, R7, and R8 were not expressed in these same samples and, as expected, the expression of R6 was responsive to the pathogen (Supplementary Fig. 6). Therefore, R6 is the only likely candidate for *Rps11* and its presence can be monitored by its expression. This conclusion is further supported by the observation that all recombinants in which expression of R6 was detected were resistant to race 1, whereas all recombinants in which expression of R6 was not detected were susceptible to race 1 (Fig. 1c and Supplementary Fig. 7). Together, these observations strongly suggest R6 is indeed *Rps11*, which is also expressed in other tissues (Fig. 1d).

The genomic region of *Rps11* between "ATG" and "TAG" producing the start and stop codons, respectively, is 14.1 kb, but its transcription start region (TSR) is located 13.1 kb upstream of the "ATG" as determined by 5′-rapid amplification of cDNA ends (5′-RACE) (Fig. 1e) and its transcription termination site (TTS) is 0.5 kb

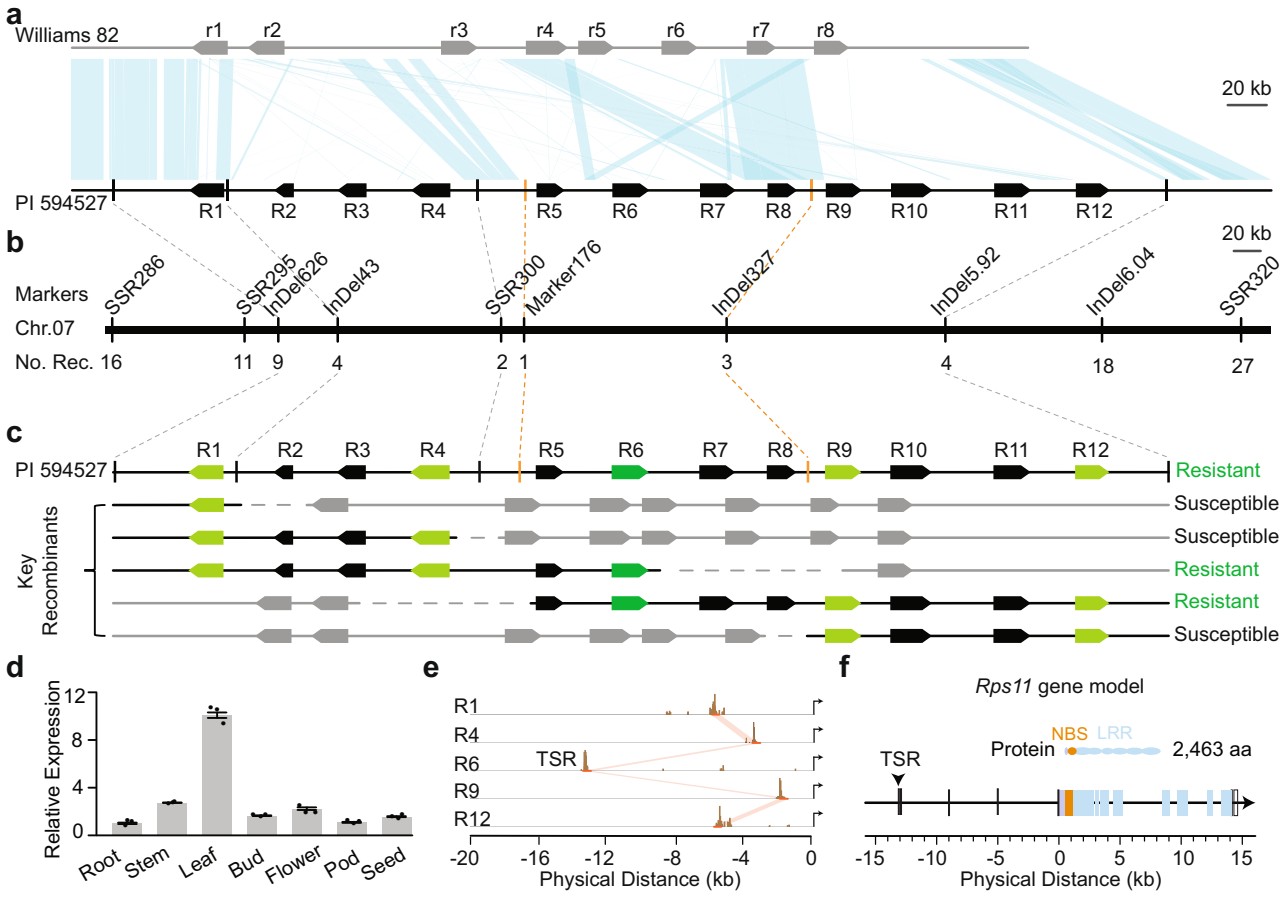

**Fig. 1 Map-based cloning of the *Rps11* locus. a** Comparison of the NLR gene clusters between Williams 82 (assembly V3.0) and PI 594527. Black boxes represent NLR genes and light-blue shades represent syntenic blocks between two genomes. **b** Physical locations of the markers used for fine mapping of the *Rps11* locus. The final mapping boundaries are highlighted by orange dashed lines. **c** Expression analyses in key recombinants by RNA-seq. Green/light-green boxes represent the five expressed NLR gene from PI 594527, black boxes represent the non-expressed NLR genes from PI 594527. Gray boxes represent the NLR genes from Williams 82. Dashed lines represent unknown genotype. The phenotype of each key recombinant is indicated as resistant or susceptible at the right side of each recombinant type. **d** Relative expression of R6 in different tissues. Values were shown as mean ± SEs of the means from three biological replicates normalized to expression of R6 in the root sample, which was set as 1.0. **e** 5′-Rapid amplification of cDNA ends (5′-RACE) performed for the 5 expressed NLR genes (R1, R4, R6, R9, and R12) to determine transcription state region (TSR). *x* axis represents the distance upstream of the first exon. Brown bars represents the 5′-RACE reads mapped to each NLR gene. Orange lines/shades show the promoter regions sharing sequence similarity. Arrows at the right side indicate the direction of the NLR genes. **f** Gene model of *Rps11* (R6). The vertical arrow points to the TSR. Orange color indicates the region encoding the NBS domain; light-blue color indicates the regions encoding the LRR domains; gray color indicates the region without a domain detected. Open boxes represent genomic fragments corresponding to 5′- and 3′-UTRs. Source data underlying Fig. 1d are provided as a Source Data file.

downstream of the "TAG," 27.7 kb of genomic DNA in total (Fig. 1f). The other expressed NLR genes including R1, R4, R9, and R12 share the conserved TSR with *Rps11*, but the distances from their TSRs to the respective "ATG" of the four genes are much shorter than that of *Rps11*, ranging from 1.8 to 5.6 kb. Although *Rps11* possesses an uncommonly long region from the "TSR" to the "ATG," this region carries three large introns and one small intron (Fig. 1f), and is primarily transcribed into a 560 bp 5′-untranslated region (5′-UTR) of mRNA. The predicted gene model of *Rps11* is validated by the RNA-seq data from PI 594527 and its coding sequence (CDS) is predicted to contain 7392 bases, encoding an NLR protein of 2463 amino acids (Supplementary Data 2).

**The resistance conferred by *Rps11* is demonstrated through transformation.** To further demonstrate the function of *Rps11*, we conducted a complementation test by developing *Rps11* transgenic lines and examining their responses to the pathogen. Ideally, *Rps11* genomic DNA would be transformed into soybean with its own promoter to best reveal the resistance conferred by *Rps11*. However, given the large size of *Rps11* from the TSR to the TTS (27.7 kb), the

unclear boundary of its promoter region upstream of the TSR, as well as anticipated difficulty in transforming such a large segment in soybean—one of the most difficult crops to transform even for DNA fragments as small as a few kb[12]—we decided to use the promoter of an *Arabidopsis* ubiquitin gene, *AtUbi3* (p*AtUbi3*), to drive *Rps11* CDS (CDS-*Rps11*) expression (see "Methods"). The p*AtUbi3*::CDS-*Rps11* construct was introduced into an elite soybean variety 93Y21, which is susceptible to *P. sojae* races 25, 31, and OH12108-06-03. As expected, the $T_2$ progeny with the transgene showed resistance to these three races at levels similar to those shown by a homozygous $F_5$ RIL (*Rps11*/*Rps11*) derived from the mapping population (Fig. 2a, b and Supplementary Fig. 8). By contrast, the non-transgenic 93Y21 (control) was susceptible to these same races (Fig. 2a, b and Supplementary Fig. 8). Also, as expected, the expression of the *Rps11*-transgene was detected in those $T_2$ progenies (Fig. 2c) and its expression level in each of the two transformation events is correlated with the level of resistance as indicated by the survival rate of inoculated seedlings (Supplementary Fig. 9).

Given that effector-triggered immunity is generally sensitive to NLR dosage, and that overexpression of functional NLR genes can

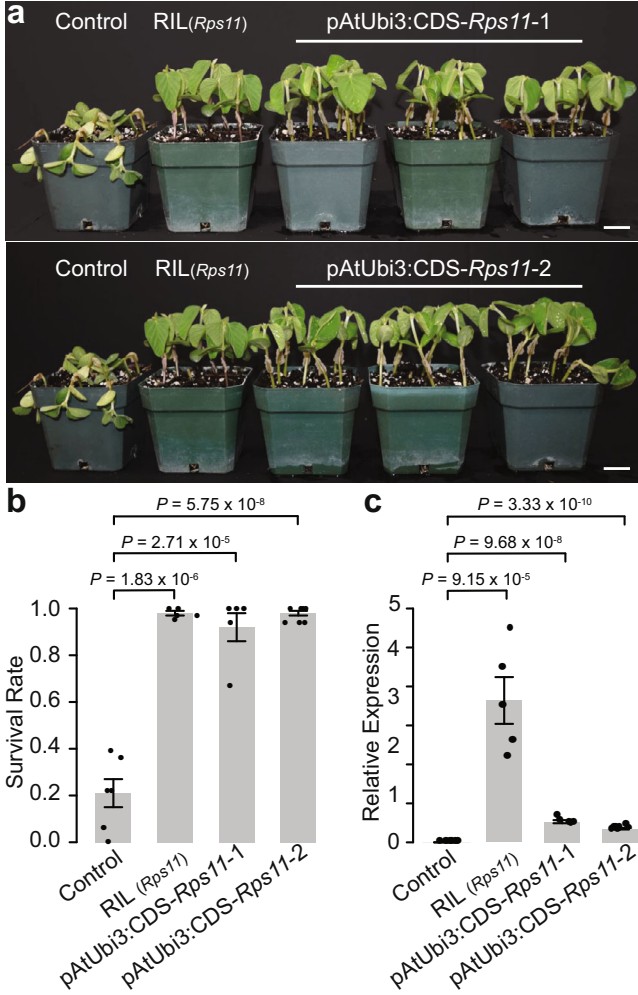

**Fig. 2 Complementation test results. a** Photographic illustration of the resistance in two independent transgenic lines. In each transgenic event, three homozygous $T_2$ families, a non-transgenic line (Control), and a $F_5$ RIL ($Rps11/Rps11$) were inoculated. Scale bars = 2.5 cm. **b** Statistics of the resistance test of homozygous $T_2$ families ($n = 5$ and 7 $T_2$ families from the two independent transgenic lines, respectively) compared with non-transgenic lines (Control, $n = 6$ non-transgenic $T_2$ families) and the $F_5$ RIL ($n = 5$ biologically independent samples) using Race 25. Data are presented as mean values ± SEM. The $Y$-axis is the survival rate after inoculation. The error bars represent SEs. The statistical significance was determined by a two-sided $t$-test and the $P$-values were shown above the plot. **c** Relative expression of the transgene (R6) compared to endo-reference gene in the two transgenic lines ($n = 5$ and 7 $T_2$ families from two independent transgenic lines, respectively), non-transgenic lines (Control, $n = 6$ non-transgenic $T_2$ families) and the $F_5$ RIL ($n = 5$ biologically independent samples). Data are presented as mean values ± SEM. The error bars represent SEs. The statistical significance was determined by a two-sided $t$-test and the $P$-values were shown above the plot. Source data underlying Fig. 2b, c are provided as a Source Data file.

result in autoimmunity[13], we wondered whether the resistance carried by the transgenic $T_2$ progeny lines could result from autoimmunity triggered by overexpression of the $Rps11$-transgene under the regulation of the $AtUbi3$ promoter. To explore this possibility, we examined the expression levels of the transgene in the $T_2$ progeny lines derived from the two transformation events. We found that the expression level of the transgenic $Rps11$ was actually lower than that of $Rps11$ in the $F_5$ RIL (Fig. 2c). Thus, the $Rps11$-transgene was not "overexpressed" at all in these transgenic lines.

Given that $Rps11$ is also highly expressed in other soybean tissues of PI 594527 (Fig. 1d), the resistance shown by the transgenic $Rps11$ reflects the function of the native $Rps11$ gene.

**$Rps11$ possesses a chimeric structure and lacks allelic copies in the soybean pan-genome.** In the PI 594527 genome assembly, a total of 512 NLR genes were annotated (Supplementary Table 2 and Supplementary Data 3). Apart from the large NLR genes in the $Rps11$ region, there are six additional large NLR genes in the genome—one located on chromosome 16 and five clustered in a region on chromosome 18. Similar to what was observed in PI 594527, large NLR genes in Williams 82 are distributed in these regions, which account for 2.7% of the 481 NLR genes annotated in the Williams 82 v2.0 genome assembly. Approximately 464 of the NLRs are predicted to produce CDSs of <5 kb per gene on average, with a mean of 2.7 kb (Supplementary Fig. 10). Compared with a typical NLR gene, the large NLR genes in PI 594527 were primarily enlarged by tandem duplications of the LRR domain (Fig. 3a).

To understand the origin and evolutionary history of these large NLR genes, we constructed a phylogenetic tree showing the relationships of all 512 NLR genes in PI 594527 using the more structurally conserved NB-ARC domains among these genes for accuracy. We found that all these large genes are phylogenetically grouped into a single clade (Fig. 3b) and share similar structures (Supplementary Fig. 4a), suggesting that they share a single origin. Comparative genomics analysis reveals that these three genomic regions hosting the large NLR genes were likely derived from two soybean whole-genome duplication (WGD) events[14]. Dot plot and phylogenetic analyses suggest that the 12 NLR genes in the $Rps11$ region were likely derived from a single copy R5, after the second WGD event occurred ~13 million years ago, through rounds of tandem duplication events, each involving one or two adjacent NLR genes (Fig. 3c–e). Apparently, the tandem duplications were also accompanied by the removal of NLR genes from the region, leading to the absence of allelic copies of most NLR genes between PI 594527 and Williams 82/Williams.

Subsequently, we annotated a total of 296 NLR genes in the genomic regions, which correspond to the ~648 kb $Rps11$ region, from the high-quality genome assemblies of 28 soybean accessions chosen to construct a pan-genome[15–18], and observed a striking copy number variation of NLR genes among these varieties, ranging from 5 to 23 copies (Fig. 4 and Supplementary Data 4). Such variation is primarily caused by unequal recombination between different NLR genes, resulting in segmental duplications, deletions, and inversions (Fig. 4, Box 1 and Box 2). The variation appears to be associated with the phylogenetic relationships among the accessions revealed by the single nucleotide polymorphism (SNP) data within the $Rps11$ region (Fig. 4). Inter-genomic comparison demonstrates that $Rps11$ is a unique gene in PI 594527 without an allelic copy in the other accessions (Supplementary Fig. 11). Nevertheless, the ~13.1 kb region from the TSR to the "ATG" of $Rps11$ was found to match the genomic sequences upstream from the predicted CDS of two adjacent NLR genes in W05, suggesting that the ~13.1 kb region of $Rps11$ was formed by fusion of such two corresponding genomic sequences in PI 594527 after its divergence from W05 (Fig. 4, Box 3). In addition, the presence/absence of the conserved TSR is likely responsible for the drastic expression variations of the NLR genes in the $Rps11$ region and "$rps11$" regions (which lack an allelic non-functional copy of $Rps11$), across the diverse soybean genomes (Fig. 4, Box 4 and Supplementary Data 4).

## Discussion

We isolated $Rps11$ conferring a broad spectrum of resistance to $P.$ $sojae$ in soybean using a gap-free sequence-based fine mapping and cloning approach, and demonstrated the origin and evolutionary

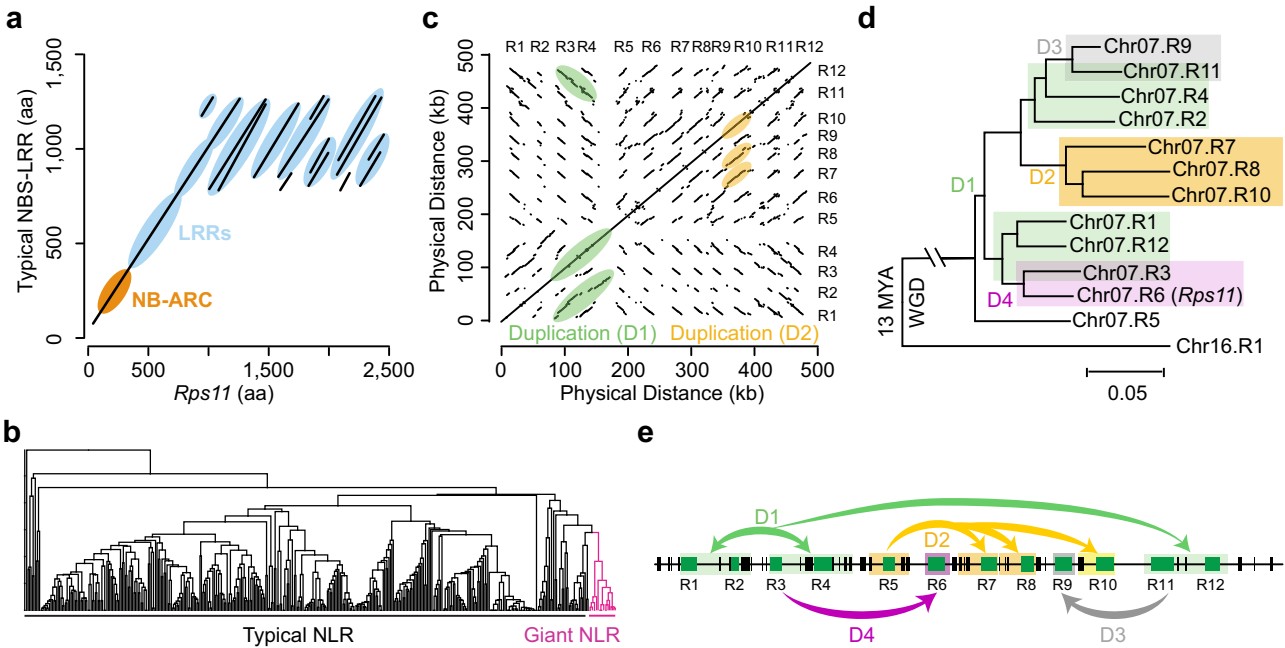

**Fig. 3 Evolutionary history of the giant NLR genes in the *Rps11* locus. a** Comparison of protein sequences between a giant NLR gene and a typical NLR gene. Lines represent the alignments between two protein sequences. Orange color highlights the NB-ARC domain region. Light-blue color highlights the LRR domain regions. **b** Phylogenetic tree of all the NLR genes in PI 594527 built using the conserved NB-ARC domain region. Giant NLR genes and typically sized NLR genes are labeled. **c** Dot plot of sequence comparison within the *Rps11* region in PI 594527. Light-green color highlights the segmental duplication of R1–R2, R3–R4, and R11–R12 (Duplication 1, D1). Light-orange color highlights the segmental duplication of R7, R8, and R10 (Duplication 2, D2). **d** Phylogenetic relationship of all the NLR genes underlying *Rps11* and the NLR gene, Chr16.R1, from its whole-genome duplication region, constructed using transcript sequence. Background colors highlight the groups produced by each duplication event. Green color highlights the duplication (D1) event of R1–R2, R3–R4, and R11–R12; orange color highlights the duplication (D2) event of R7, R8, and R8; gray color highlights the duplication (D3) event between R9 and R11; purple color highlights the duplication (D4) event between R3 and R6. **e** Illustration of the evolutionary history of the giant NLR gene cluster in the *Rps11* region. Green boxes represent NLR genes and black boxes represent predicted non-NLR genes. Green arrows indicate the duplication (D1) event of R1–R2, R3–R4, and R11–R12; orange arrows indicates the duplication (D2) event of R7, R8, and R8; gray arrow indicates the duplication (D3) event between R9 and R11; purple arrow indicates the duplication (D4) event between R3 and R6.

dynamics of a NLR gene cluster. Given such a magnitude of structural and gene copy number variations of the NLR gene cluster and the lack of allelic copies of the *Rps11* locus in the soybean pan-genome, the sizable mapping population and the high quality of genome assemblies of both parental lines have been essential for successful isolation of *Rps11*.

A number of studies have revealed structural and copy number changes of NLR genes underlying resistance to plant pathogens. In maize, the *Rp1* locus conferring rust resistance was found to be flanked by additional NLR genes and shaped by a number of unequal intergenic recombination events, particularly in LRR domains, which generated a variety of mutations, some of which were able to alter race-specific resistance[19,20]. In sorghum, the *Pc* locus conferring resistance to the fungal pathogen *Periconia circinate* was found to be an NLR gene flanked by additional NLR genes and the resistance of the locus is frequently overcome by site-directed unequal recombination between the flanking NLR genes[21]. A species wide record of NLR genes (called pan-NLRome) was recently reported in *Arabidopsis*, providing a panoramic view of important NLR genes in the model plant species[22]. More recently, 26 diverse maize genomes were sequenced, allowing for a glimpse of the haplotypic variation in the *rp1* region[23]. The region exhibited drastic inter-genomic copy number variation of *rp1*, but a subset of these genomes still have gaps in this region, and the structure and evolutionary processes of the *rp1* region and many other NLR clusters in the genomes remain to be analyzed. The detailed comparative analyses of the gap-free *Rps11* regions sheds light on the evolutionary plasticity and consequences of resistance genes such as gain, loss, and reinforcement of resistance.

Previously, two small NLR genes have been reported to be responsible for Rps resistance conferred by the *Rps1-k* locus in Williams 82[24,25]. Intriguingly, neither of the two genes can be found in any versions/sources of the Williams 82 genome assemblies including unassembled contigs (soybase.org; 9). Thus, the candidate gene(s) of *Rps1-k* remains to be validated. In addition, *Rps1-k* has become ineffective to many of the newly evolved *P. sojae* races[8]. Therefore, the source of resistance conferred by *Rps11* is highly desirable. The LRR domains in NLR genes are involved in determining the plant's ability to recognize specific pathogen effectors[26,27], but whether the expanded LRR domain in *Rps11* is responsible for such a broad resistance spectrum remains to be tested. Introns in the 5′-UTR have been detected in many eukaryotic genes and recent studies have demonstrated that they can regulate gene expression at multiple levels[28,29], but whether and how the three large introns in the 5′-UTR of the *Rps11* transcript affect its biological function remains unclear. The unique structural features of *Rps11* make this NLR a suitable model for investigating molecular mechanisms underlying the broadness of resistance spectrum and for developing strategies such as engineering of NLR genes for resistance enhancement. In addition to the functional and evolutionary insights learned from the *Rps11* gene cluster, this study also provides *Rps11*-based unique markers for precise selection of the gene and will accelerate the deployment of the gene for soybean protection

## Methods

**Plant materials and resistance evaluation**. The mapping populations were generated from an initial cross between PI 594527 and Williams. In total, 17,050

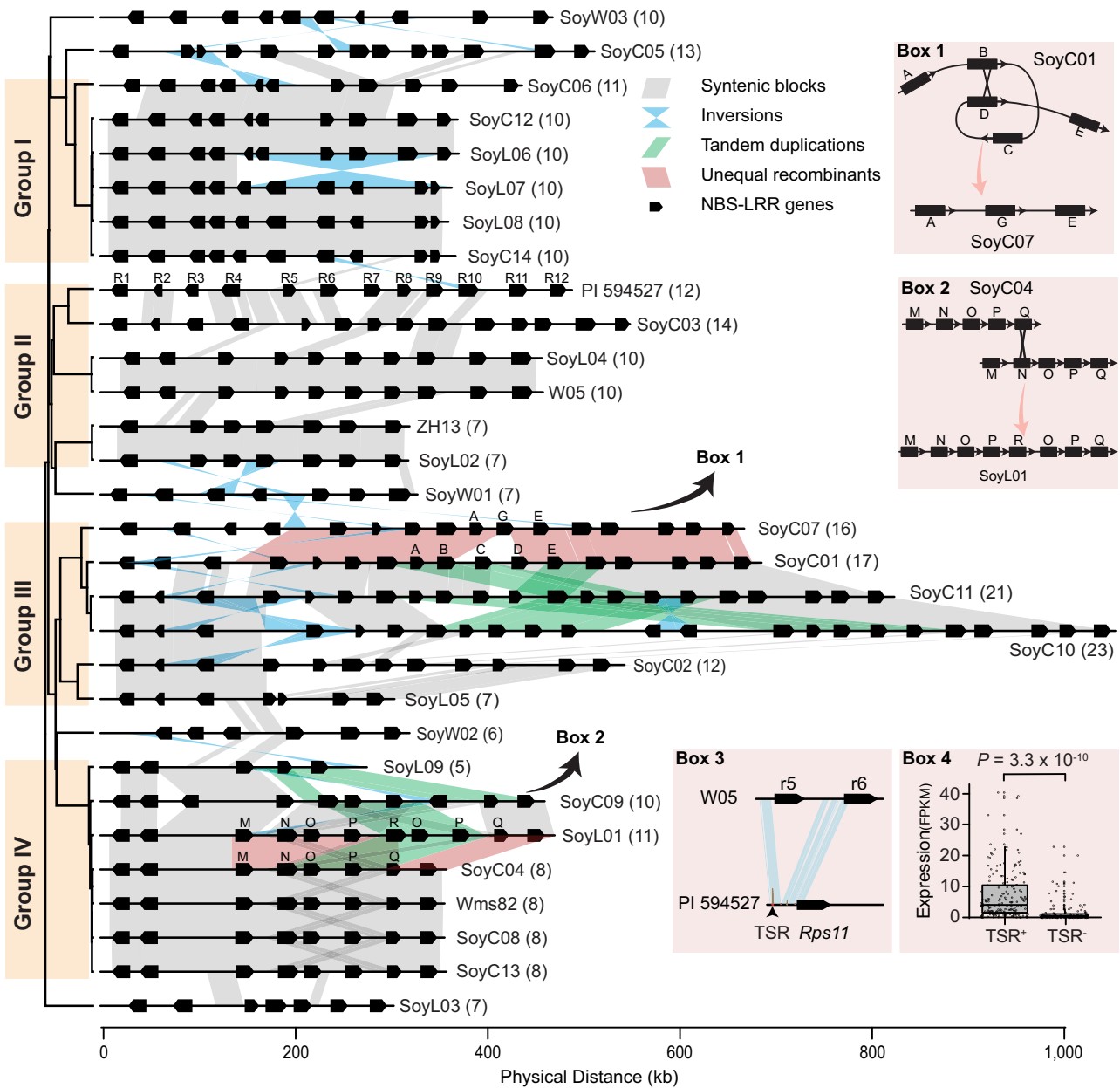

**Fig. 4 Diversification of the NLR gene cluster across 30 diverse soybean genomes.** The phylogenetic tree on the left side was built using SNP data with the *Rps11* region. Orange color highlights the four major haplotype groups. Each black box represents an NLR gene. Gray shades represent syntenic blocks among genomes. Light-blue highlight inversion events. Green highlight tandem duplication events. Light-red highlight potential unequal recombination events. Box 1 and Box 2 exemplify a deletion and a duplication event induced by unequal recombination, respectively. The name of each genome was labeled at the right side of each cluster and the numbers in parentheses are the total copy number of NLR genes in each genome at the *Rps11* corresponding region. Box 3 shows the sequence comparison of the promoter regions of *Rps11* and that of W05.r5 and W05.r6. Light-blue shades represent syntenic blocks. Box 4 shows the comparison of the expression levels of NLR genes with TSR (TSR+, *n* = 187 NLR genes from 30 soybean accessions) and without TSR (TSR−, *n* = 129 NLR genes from 30 soybean accessions) in the *Rps11* region and "*rps11*" regions across all the genomes. Horizontal lines indicate the medians and the boxes represent the interquartile range (IQR). The whiskers represent the range of 1.5 times IQR and dots beyond the whiskers are outlier values. The statistical significance was determined by a two-sided *t*-test and *P*-value is shown above the plot. Source data underlying Box 4 are provided as a Source Data file.

F$_4$ plants derived from heterozygous F$_3$ individuals were screened for identification of recombinants. Only the recombinants with heterozygous genotype at one boundary marker and homozygous Williams genotype at the other boundary marker were used for fine mapping, because the two expected phenotypes could be accurately distinguished. About 30 seedlings from each line were inoculated with Race 1 of *P. sojae* using the hypocotyl inoculation method[3,30]. Lines with <25% of progenies surviving after inoculation were classified as susceptible; recombinants with 25–75% of progenies surviving were classified as segregating, lines with >75% of progenies surviving after inoculation were classified as completely resistant.

**Genotyping the recombinants.** SSR markers and insertion/deletion markers were identified and designed based on re-sequencing data of the two parental lines. Marker_176kb was a dominant marker that could only be amplified from the donor line. Only the markers with a unique hit at the *Rps11* region were used for fine mapping. Competitive allele-specific PCR markers were also used to identify and genotype the recombinants. All DNA markers used in this study were ordered from Integrated DNA Technologies (Coralville, IA, USA) and are listed in Supplementary Table 3.

**Long- and short-read genome sequencing.** Long-read data were generated using the Pacific BioSciences (Menlo Park, CA, USA) Sequel platform at Corteva Agriscience™

(Johnston, IA, USA). Eight SMRT cells were performed with 10 h movies and v6 chemistry. Raw subreads were filtered to a minimum of 12 kb generating $77 \times$ genome coverage. The raw subread N50 length was 28.9 kb. Linked short-read data were generated by sequencing 10 × Genomics (Pleasanton, California) Chromium libraries at Corteva Agriscience™ on the Illumina (San Diego, California) HiSeq2500 platform in a PE151 configuration. The coverage depth and mean molecule length for the Chromium library were $45.2 \times$ and 93.8 kb, respectively.

**Genome assembly and sequence polishing.** Canu[31] v1.8 (https://github.com/marbl/canu) was used to self-correct the raw subreads and to assemble the corrected reads into contigs. The following changes were made to the default parameters: correctedErrorRate = 0.065, corMhapSensitivity = normal, and ovlMerDistinct = 0.99. A minimum contig length of 30 kb was applied. Additional sequence polishing was performed by aligning raw PacBio subreads to the contig assembly using pbmm2 v0.12.0 (https://github.com/PacificBiosciences/pbmm2) and applying the Arrow algorithm from the Genomic Consensus package v2.3.2 (https://github.com/PacificBiosciences/GenomicConsensus) to identify and correct the remaining consensus errors in the contigs. These tools were acquired from pbbioconda (https://github.com/PacificBiosciences/pbbioconda). The consensus sequence accuracy was further enhanced by complementing the long-read contig assembly with Chromium linked short reads. Chromium data sets were aligned to contigs using Long Ranger v2.2.2. The sequence assembly polishing tool Pilon[32] v1.22 (https://github.com/broadinstitute/pilon) was used to correct individual base errors and small indels from the Chromium data aligned to the contigs using the "–fix bases –minmq 30" parameters.

**Creating genome maps.** Genome maps were generated in the Bionano Saphyr platform at Corteva Agriscience™ using the Direct Label and Stain approach[33]. Nuclear DNA was isolated from leaf tissue using a modified version of the Bionano Prep™ Plant Tissue DNA Isolation protocol (https://Bionanogenomics.com/wp-content/uploads/2017/01/30068-Bionano-Prep-Plant-Tissue-DNA-Isolation-Protocol.pdf), which did not include a gradient centrifugation step. DLE-1-labeled molecule data were filtered to create a data set with a molecule N50 of 441 kb and 267× coverage. The resulting genome maps were filtered to remove coverage and length outliers. The final genome map assembly consisted of 43 maps with a genome map N50 of 26.7 Mb and a total map length of 985 Mb. This data set was assembled via the Bionano Genomics Access software platform (Solve3.2.2_08222018) with the configuration file optArguments_nonhaplotype_noES_noCut_DLE1_saphyr.xml.

**Hybrid scaffolding of genome maps with sequence contigs.** Hybrid scaffolds were generated from the polished contigs and the Bionano genome maps using the Bionano Genomics Access software (Solve3.3_10252018) and the DLE-1 configuration file hybridScaffold_DLE1_config.xml. In addition to auto-conflict resolution performed by the software, manual curation was performed to resolve overlapping and embedded contigs by providing additional "Conflict resolutions" and re-running the hybrid scaffolding. In the final product, the assembly had 43 hybrid scaffolds (Scaffold N50 = 26.4 Mb, Total scaffold length = 978.1 Mb) with 229 leftover contigs that were not scaffolded with a combined length of 21.3 Mb.

**Building chromosome-scale pseudomolecules.** A reference-based approach was feasible to create chromosome-scale pseudomolecules using the *Glycine max* Wm82.a2.v1 reference assembly (https://phytozome.jgi.doe.gov/), because only an average of 2.15 scaffolds per chromosome needed to be placed. To map hybrid scaffolds to the reference, each scaffold was chunked into 100 bp fragments and then aligned to the reference genome using minimap2[34] v2.10 (https://github.com/lh3/minimap2). Then, order and orientation of scaffolds were determined by the coordinates of the scaffold-chunk reads relative to the reference sequence. All scaffolds were able to be placed using this method. Leftover unscaffolded contigs were concatenated with 100 bp N-gaps and assigned to Chr00.

**NLR gene annotation and expression analysis.** NLR genes were annotated using NLR-Annotator[35]. RNA samples were extracted from mixed stem tissues from seedlings of each key recombinants using RNeasy Plant Mini Kit (catalog number 74904, Qiagen) and were treated with RNase-Free DNase Set (catalog number 79254, Qiagen) to remove DNA. RNA-seq was performed by Purdue Genomic Core Facility (https://www.purdue.edu/hla/sites/genomics/). Public RNA-seq data were collected and extracted using sra-toolkit v2.11.0 (https://trace.ncbi.nlm.nih.gov/Traces/sra/sra.cgi?view=software). RNA-seq data were mapped to the genome of the donor line using STAR v2.7.9a[36] and expression was calculated based on the number of reads mapped to each NLR gene. Gene expression profiling by quantitative real-time PCR was performed using soybean ATP-binding cassette transporter gene (dubbed *Cons4*) as an endogenous control following a protocol described previously[37].

**Phylogenetic analysis.** Phylogenetic trees in this study were constructed using the Minimum Evolution method[38] in MEGA7[39]. Only NB-ARC region was used in Fig. 3b, because the NB-ARC domain is relatively conserved among all NLR genes. The LRR regions are highly divergent in both sequence diversity and copy number variations; thus, it is difficult to align the LRR regions. Both the NB-ARC and LRR regions were

used in Fig. 3c, because these NLR genes share the same origin and high sequence similarity.

**Plasmid construction and transformation.** To make the overexpression construct for the *Rps11* candidate gene *R6*, the CDS of R6 was synthesized by Genscript as three fragments and assembled with *AtUbi3* promoter and Gateway ATT sites by homologous recombination in yeast to make a Gateway entry vector[40]. This was recombined into a Gateway destination vector by attL-attR recombination reaction using Gateway Technology with Clonase II (25–0749, Invitrogen) for transformation into *Ochrobactrum*. *Ochrobactrum*-mediated soybean embryonic axis transformation was done as previously described (US20180216123A1; WO2020/005933A1; and WO2020/092494A1). Mature dry seeds of soybean cultivar 93Y21 were disinfected using chlorine gas and imbibed on semi-solid medium containing 5 g/L sucrose and 6 g/L agar at room temperature in the dark. After an overnight incubation, the seed was soaked in distilled water for an additional 3–4 h at room temperature in the dark. Intact embryonic axis explants were isolated and transferred to the deep plate with 15 mL of *Ochrobactrum haywardense* H1-8 suspension (optical density 0.5 at 600 nm) in infection medium composed of 1/10X Gamborg B5 basal medium, 30 g/L sucrose, 20 mM MES, 0.25 mg/L GA3, 1.67 mg/L BAP, 200 μM acetosyringone, and 1 mM dithiothreitol in pH 5.4. The plates were sealed with parafilm ("Parafilm M" VWR catalog number 52858), then sonicated (Sonicator-VWR model 50 T) for 30 s. After sonication, embryonic axis explants were transferred to a single layer of autoclaved sterile filter paper (VWR#415/catalog number 28320-020). The plates were sealed with Micropore tape (catalog number 1530-0, 3M, St. Paul, MN) and incubated under dim light (5–10 μE/m²/s, cool white fluorescent lamps) for 16 h at 21 °C for 3 days.

After co-cultivation, the base of each embryonic axis was embedded in shoot induction medium (R7100, PhytoTech Labs) containing 30 g/L sucrose, 6 g/L agar, and 25 mg/L spectinomycin (S742, PhytoTech Labs) as a selectable agent and 500 mg/L cefotaxime (GoldBio, St. Louis, MO, USA) in pH 5.7. Shoot induction was carried out at 26 °C with a photoperiod of 18 h and a light intensity of 40–70 μE/m²/s. After 4–6 weeks in selection medium, the spectinomycin-resistant shoots were cut and transferred to ½ strength MS rooting medium (M404, PhytoTech Labs) containing 15 g/L sucrose, 6 g/L agar, 10 mg/L spectinomycin, and 250 mg/L cefotaxime for further shoot and root elongations.

Marker-free transgenic soybean plants were generated by the Cre-lox site-specific recombination system using heat shock treatment. For heat shock treatment of soybean, 2–4 cm T0 plantlets with roots in 100 × 25 mm Petri dishes on spectinomycin free-rooting medium were transferred into a Percival incubator (Percival Scientific, Perry, IA, USA) at 45 °C, 70% humidity for 2 h in the dark. After the heat shock treatment, T0 plantlets were transferred to moistened Berger BM2 soil (Berger, Saint-Modeste, QC, Canada) and kept enclosed in clear plastic tray boxes in a Percival incubator at 26 °C with a 16 h photoperiod at 250–350 μE/m²/s. Leaf punch samples were collected for molecular analyses from newer growth 2 weeks after acclimatization of T0 events. Hardened plantlets were potted in 2 gal pots containing moistened SunGro 702 and grown to maturity for harvest in a greenhouse. The presence of the construct in the transgenic plants was confirmed by PCR with primers specific to the cloning vector and expression analysis of R6 in the transgenic plants by qPCR.

**5′-RACEs and TSR analysis.** Next, 5′-RACE was performed using the Gene Racer Kit (Invitrogen, MA) following the manufacturer's protocol. The cDNA sample from PI 594527 was amplified by nested PCR, with reverse primers designed at the conserved regions shared by R1, R4, R6, R9, and R12. PCR products were sequenced by WideSeq (https://www.purdue.edu/hla/sites/genomics/wideseq-2/) and only the reads containing the 5′-ends were selected and mapped to the PI 594527 genome.

The regions containing 5′-ends upstream of each expressed NLR gene were defined as the TSR. To determine the presence/absence of the TSR in all the NLRs in the *Rps11* and *rps11* regions, we blasted the TSR sequence from R6 against the 20 kb sequence from each NLR. TSR$^+$ indicates the presence of a blast hit at upstream of an NLR; TSR$^-$ indicates no blast hit was found at the upstream of an NLR. Expression levels of the NLR genes were calculated using the RNA-seq data from each genome.

**Reporting summary.** Further information on research design is available in the Nature Research Reporting Summary linked to this article.

## Data availability
All the raw sequence data and the genome assembly of PI 594527 generated from this article have been deposited in the NCBI database under BioProject PRJNA718574. Other genome assemblies and RNA-seq data were generated by previous studies and are publicly available[9,15–18]. The *Rps11* donor line (PI 594527) is available from the USDA germplasm collection (https://npgsweb.ars-grin.gov/). The transgenic materials described in this study are protected by the patent specified in "Competing interests" section and may also be subject to other intellectual property rights and regulatory and stewardship constraints. Corteva Agriscience™ reserves the right to require a requester of such materials to enter into a non-disclosure agreement and a material transfer agreement in order to receive the materials. The use of the materials will be limited to non-commercial research uses only. Please contact R.A. (rajat.aggarwal@corteva.com) or J.M. (maj@purdue.edu) regarding the transgenic materials, and requests will be responded within 60 days. Source data are provided with this paper.

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

## Acknowledgements

This work was mainly supported by Corteva Agriscience™. We thank Jon Massman and John Woodward for their leadership in the Purdue University-Corteva Agriscience project collaboration. We thank Ajit Nott, Lyudmila Sidorenko, Abhijit Sanyal, Jon Allen, and Tyler Engelhart for their technical support in transformation, sequence analysis, resistance screening, and greenhouse coordination. We also thank Dr. Xianzhong Feng for providing the Williams 82 assembly v3.0. Publication of this article was funded in part by Purdue University Libraries Open Access Publishing Fund.

## Author contributions

J. Ma, R.A., O.C., J. Munkvold, A.B. and J.T. designed the research. W.W., L.C., K.F., J. Myrvold, J.B., X.W., V.L., C.B.C., T.J.F., D.O., A.H., M.M.v.D. and G.C. performed the experiments. W.W., L.C., K.F. and X.W. analyzed the data. J. Ma and W.W. wrote the manuscript.

## Competing interests

Patent application (including 10,995,377) on the gene sequence, associated markers, and transgenic materials described in this study has been filed by Corteva Agriscience™.
