## [Peer Review File · Nature Communications]

A giant NLR gene confers broad-spectrum resistance to
Phytophthora sojae in soybeanReviewers' Comments:

Reviewer #1:

Remarks to the Author:

The authors have discovered a fascinating example of plant immune system evolution and diversification. The discovered example is novel in its extreme nature and its clarity - it is a beautiful example of R gene/locus evolution concepts that have been around and demonstrated in this discipline for 20 years - but it is a better example. This is due to a combination of finding an R locus with interesting behavior, having cutting-edge capacity to assemble and analyze gap-free genomic sequences for the loci from multiple soybean accessions, and having the high-level insight to interpret the findings and highlight points of interest. I agree with their identification of R6 as the causal gene. The authors provide an example that, by careful documentation of the facts, expands our sense of what can happen with natural or engineered plant immune receptor evolution. I agree with the Discussion sentence that "few NLR clusters harboring known resistance loci have been fully sequenced and compared at the population level, this study sheds light on the evolutionary plasticity and consequences of resistance genes such as gain, loss, and reinforcement of resistance." The authors supply a highly citable example to use in the future when students are taught about plant immune systems or when authors discuss the plasticity of R gene loci. My main concern is with a few important details, regarding the abstract and regarding availability of the biological materials. I also regret that the authors have pursued a short-manuscript format that prevents more detailed explanation of their Results and more extensive Discussion. But I support acceptance and publication of this manuscript.

Specific Comments:

Lines 17-31: The abstract is overwrought in multiple places. Adjectives like "urgent" and "extreme" should be deleted. Or if the authors insist on decorative adjectives, choices like "pressing" and "strong" would be more appropriate (but best to just delete those adjectives). Also: *Phytophthora sojae* is not "the most" destructive soil-borne disease of soybean (that would be SCN - see ref. 4 Wrather, or more modern sources). *P. sojae* is "one of the most destructive" or "a destructive". Line 29-30 should say: "Rps11 thus exemplifies..." (not "Our study exemplifies")(lines 30-31 could say: "...and the present study will accelerate..."). Ironically, the word "drastic" on line 27 seems to me to be an accurate scientific descriptor of the Rps11 locus, and its use is appropriate in the abstract. But for example, do not call Rps11 "extreme broad spectrum" in the abstract (or on line 143 or 159) if it fails against 20% of the tested isolates.

Abstract Line 26: "...that presumably explains such broadness of the resistance spectrum." This is a red-flag sentence - draws attention to the fact that the authors have not demonstrated this point. Could say "...that may contribute to the broad resistance spectrum."? It is OK that this point is not nailed down yet for this paper - that may require a few more years of study. The present study still makes a valuable contribution.

Lines 406-422

For example, regarding Extended Data Fig. 9, which is central to the value of the manuscript: to what extent will the discovered materials be available? Other researchers should absolutely have free access to the DNA sequence data for the Rps11 locus - it sounds as if that will universally be the case (but check to be sure). But researchers in the plant sciences typically expect access to the germplasm - for research use. This is a tricky area, and it is good for fellow academics and journals to promote academic/industrial interaction by allowing Corteva to protect their germplasm (i.e., not share it if they don't trust that the germplasm will be used only for further research on the narrow topic of the present publication). Corteva has an admirable track record of actually participating in/contributing to the public/academic knowledge sector, including allowing researchers to access germplasm. Their participation may dry up if they are forced to pass out their elite germplasm. But the editors of Nature Communications are cautioned to carefully assess all of lines 406-422. Note for example line 409 "may be available" is clarified in subsequent lines to mean "may not be available." I am fine with lines

406-422 as written. Just want to be sure the journal is OK with this.

Lines 54-61:

The resistance spectrum of the Rps11 haplotype from PI 594527 is not very clear from Supplemental Tables 1 and 2. Those tables are appropriate as-is; they describe experimental results. But please, without doing any more experiments, the authors should please supply readers with any additional information they have available on the race type of more of the isolates used in the experiments described in those Supplemental Tables. We understand that race-typing systems are always incomplete, but please tell us more about other Rps haplotypes upon which the isolates from Supplemental Table 1 are virulent. And can the authors comment more on the 20% of isolates from Supplemental Table 2 that were virulent on the Rps11 haplotype from PI 594527? (for example, do they represent new or old isolates, an edited set of common or uncommon isolates, etc.).

Line 80:

The authors might emphasize for readers here that R1-R12 are not close homologs of Rps11 locus r1-r8 genes from Williams 82. Then make the statement about synteny in the following sentence

Lines 143-162:

If space is available, a much longer Discussion section would be beneficial. Even if the journal demands brevity, the authors might still sneak in another sentence or two about the diversity-generation capacity of this type of R gene locus, possibly citing maize Rp1 or other published examples of high-diversification loci, and/or a review that does a strong job on that topic. It would be good to comment that NLR structure/function studies have recently advanced dramatically (see Science/Nature 2021 papers), and note that analysis of (extended-LRR) R6 protein multimer structure is of future interest. It would be nice to further emphasize for readers the elevated potential of highly recombinogenic R loci to generate immune diversity, and the possible ways that technologists might harness that. But it also bears note that Rps11 was not previously landed upon by other soybean geneticists (for *P. sojae* or other diseases).... A much longer Discussion is merited.

Minor Points:

The primary literature reference choices are appropriate, but the excellent reviews/chapters that are cited are all a bit dated - is there at least one more recent source that can be used?

Please reference the literature more and/or speculate a bit about the functional purpose of the unusually long 13 kb 5'UTR.

Methods should describe Fig. 4 box 4 presence absence was determined (how was a TSR defined? what allowed classification as TSR-?)

Reviewer #2:

Remarks to the Author:

Review: A giant chimeric NLR gene confers extreme broad-spectrum resistance 1 to a plant pathogen

Summary Review: This research group has used the traditional map-based cloning procedure to identify a single NLR gene within a cluster of NLR genes that provides resistance to a large set of *Phytophthora sojae* isolates. While the cluster was shown to provide resistance to 127 isolates, within that cluster, one gene, R6, was shown to provide resistance to three isolates by analyses of transgenic plants with the R6 gene. This appears to be a breakthrough of importance to soybean production because *P. sojae* is a major plant pathogen on the crop.

The group then goes on to discuss the evolution of the cluster where a single (or two) genes underwent tandem duplication to evolve the cluster found in the resistant genotype. This cluster was further studied in the assembled genomes of 28 other genotypes where it was found the cluster shows wide variation in NLR number and structure. I would ask, have the evolution of other clusters been described to the detail found here. If not, that should be stated. If so, a discussion of how this cluster and other clusters evolved should be included in the narrative.

Research Approach: All of the experimental protocols were appropriate to draw the conclusions reported in the manuscript. The procedures are described in enough detail for others to understand the conclusions, repeat the work, and use the outcome in applied crop improvement.

General: The authors should consider using race or isolate not both somewhat interchangeably.

Line 36: "subsequent Irish diaspora"; diaspora by itself generally means the movement of Jewish peoples

Line 41: "soybean genomic regions"

Line 65: What information suggest Rps11 "had not been fully assembled" in Williams 82?

Line 72: What is a Rps11/rps11 region. It would be better to say "Rps11 and rps11 regions".

Line 75: please discuss why these NLRs are considered "large"; nucleotide length compared to what? Size of proteins compared to what?

Line 117: What phylogenetic approach was used? Why was only the NB-ARC region used? This is not presented in the M&M or Fig. 2 legend.

Line 119: "...were derived..." seems incorrect. The authors provide evidence that a single (or two) genes underwent tandem duplications to evolve the cluster. Importantly this region is distinct from the NLR cluster in the susceptible region. The team should reconsider this issue during revision.

Line 128: susceptible accessions?? If so, mention here.

Line 139: What does Rps11/"rps11" mean?

Line 212: What assembly (V2.0 or 3.0) of Williams 82 is being displayed.

Line 222: Relative expression of what?

Line 255: What does D1 - D4 mean?

Line 279: What does "one side heterozygous and the other 279 side homozygous Williams genotype" mean. I have not heard that terminology.

Lines 414 and 416: Who is the third party that should be contacted? Researchers need to know that to confirm the work here.

Reviewer #3:

Remarks to the Author:

The manuscript by Weidong Wang et al. isolated a resistant gene Rps11 in soybean by a gap-free sequence-based fine mapping and cloning approach. Rps11 is a 27.7-kb nucleotide-binding site-

leucine-rich repeat gene which confers broad-spectrum resistance to *Phytophthora sojae* isolates. And Rps11 is located in a genomic region harboring a cluster of unusually large NLR genes. A comprehensive set of evolution data show that the NLR gene cluster exhibits drastic structural diversification caused by duplications and recombination. Overall, the data appear to be analyzed appropriately with one or two exceptions detailed below.

1. This conclusion that R6 in PI 594527 is Rps11 is not justified. R6-transgene showed resistance to all tested three races. Constitutive promoter (ubiquitin promoter) was used to express R6, an NLR gene, in an elite soybean variety 93Y21, which could have led to artificial resistance to *P. sojae* races.
2. The novelty of this study was mentioned in the abstract "Our study thus exemplifies innovative evolution of NLR genes and NLR gene clusters and will accelerate the deployment of Rps11 for soybean protection." but no further discussion of this point are mentioned in discussion section. Actually, the similar evolution of NLR genes and NLR gene clusters have been mentioned in other plants.
3. The title is a bit exaggerated, such as "giant", "chimeric", "extreme", "a plant pathogen". It should be expressed in more precise terms.
4. Statistical significance should be presented in Fig. 1d, Fig. 2b, Fig. 2c, Extended Data Fig. 3b and Extended Data Fig. 7a.
5. More detailed information should be provided in this manuscript, such as Rps11 sequence, isolates collected from Indiana and so on.
6. R6 expression of a F5 RIL should be presented in Fig. 2c as a control.

RESPONSES to REVIEWERS' COMMENTS

Reviewer #1 (Remarks to the Author):

The authors have discovered a fascinating example of plant immune system evolution and diversification. The discovered example is novel in its extreme nature and its clarity - it is a beautiful example of R gene/locus evolution concepts that have been around and demonstrated in this discipline for 20 years - but it is a better example. This is due to a combination of finding an R locus with interesting behavior, having cutting-edge capacity to assemble and analyze gap-free genomic sequences for the loci from multiple soybean accessions, and having the high-level insight to interpret the findings and highlight points of interest. I agree with their identification of R6 as the causal gene. The authors provide an example that, by careful documentation of the facts, expands our sense of what can happen with natural or engineered plant immune receptor evolution. I agree with the Discussion sentence that "few NLR clusters harboring known resistance loci have been fully sequenced and compared at the population level, this study sheds light on the evolutionary plasticity and consequences of resistance genes such as gain, loss, and reinforcement of resistance." The authors supply a highly citable example to use in the future when students are taught about plant immune systems or when authors discuss the plasticity of R gene loci. My main concern is with a few important details, regarding the abstract and regarding availability of the biological materials. I also regret that the authors have pursued a short-manuscript format that prevents more detailed explanation of their Results and more extensive Discussion. But I support acceptance and publication of this manuscript.

RESPONSE #1.1: We are grateful to the reviewer for taking the time to assess our manuscript and for providing a set of insightful and constructive comments. This manuscript was directly transferred from another *Nature* journal to *Nature communications*, and the original manuscript is relatively short and brief. Per the reviewer's suggestions, we have now expanded the Results and Discussion sections quite a bit. Nevertheless, *Nature Communications* requires even a shorter Abstract (150 words) than the other journal does (200 words), so we have shortened the Abstract in this revision. But we hope new Abstract summarizes our work even better than the previous one. Regarding the availability of the biological materials, please see the **RESPONSE #1.4**.

Specific Comments:

Lines 17-31: The abstract is overwrought in multiple places. Adjectives like "urgent" and "extreme" should be deleted. Or if the authors insist on decorative adjectives, choices like "pressing" and "strong" would be more appropriate (but best to just delete those adjectives). Also: *Phytophthora sojae* is not "the most" destructive soil-borne disease of soybean (that would be SCN - see ref. 4 Wrather, or more modern sources). *P. sojae* is "one of the most destructive" or "a destructive". Line 29-30 should say: "Rps11 thus exemplifies..." (not "Our study exemplifies") (lines 30-31 could say: "...and the present study will accelerate..."). Ironically, the word "drastic" on line 27 seems to me to be an accurate scientific descriptor of the Rps11 locus, and its use is appropriate in the abstract. But for example, do not call Rps11 "extreme broad spectrum" in the abstract (or on line 143 or 159) if it fails against 20% of the tested isolates.

RESPONSE #1.2: We thank the reviewer for pointing these out. We have made all suggested changes.

Abstract Line 26: "...that presumably explains such broadness of the resistance spectrum." This is a red-flag sentence - draws attention to the fact that the authors have not demonstrated this point. Could say "...that may contribute to the broad resistance spectrum."? It is OK that this point is not nailed down yet for this paper - that may require a few more years of study. The present study still makes a valuable contribution.

RESPONSE #1.3: Thanks. We have changed this phrase as suggested. Please see Line 27.

Lines 406-422. For example, regarding Extended Data Fig. 9, which is central to the value of the manuscript: to what extent will the discovered materials be available? Other researchers should absolutely have free access to the DNA sequence data for the *Rps11* locus - it sounds as if that will universally be the case (but check to be sure). But researchers in the plant sciences typically expect access to the germplasm - for research use. This is a tricky area, and it is good for fellow academics and journals to promote academic/industrial interaction by allowing Corteva to protect their germplasm (i.e., not share it if they don't trust that the germplasm will be used only for further research on the narrow topic of the present publication). Corteva has an admirable track record of actually participating in/contributing to the public/academic knowledge sector, including allowing researchers to access germplasm. Their participation may dry up if they are forced to pass out their elite germplasm. But the editors of Nature Communications are cautioned to carefully assess all of lines 406-422. Note for example line 409 "may be available" is clarified in subsequent lines to mean "may not be available." I am fine with lines 406-422 as written. Just want to be sure the journal is OK with this.

RESPONSE #1.4:

For data availability, all the raw sequencing data and genome assembly generated from this study have been deposited into NCBI database (BioProject PRJNA718574) and are publicly available to the community. All other genome assemblies used for the comparative genomics analyses have been described in previous publications and are publicly available (Please see Lines 409-411 and cited references). Regarding germplasm described in this study, it is a general statement for all publications involving Corteva germplasm, some of which may be associated with a third party of genetic element and thus may need specific agreements. Nevertheless, the *Rps11* donor line is freely available and others such as lines from the mapping population need regular MTAs.

Lines 54-61: The resistance spectrum of the *Rps11* haplotype from PI 594527 is not very clear from Supplemental Tables 1 and 2. Those tables are appropriate as-is; they describe experimental results. But please, without doing any more experiments, the authors should please supply readers with any additional information they have available on the race type of more of the isolates used in the experiments described in those Supplemental Tables. We understand that race-typing systems are always incomplete, but please tell us more about other *Rps* haplotypes upon which the isolates from Supplemental Table 1 are virulent. And can the authors comment

more on the 20% of isolates from Supplemental Table 2 that were virulent on the Rps11 haplotype from PI 594527? (for example, do they represent new or old isolates, an edited set of common or uncommon isolates, etc.).

RESPONSE #1.5: Thanks for the comments. Besides the set of known *P. sojae* races, the majority of isolates recently collected have not been fully characterized with a set of differentials each carrying a different Rps gene/allele yet. The co-author Guohong Cai at USDA-ARS is working on that through a separate project, and he plans to publish such data separately in the future. We have now added a new supplemental figure (Supplementary Fig. 2) to demonstrate the geographic distribution of the isolates as an indicator of their representation.

Line 80: The authors might emphasize for readers here that R1-R12 are not close homologs of Rps11 locus r1-r8 genes from Williams 82. Then make the statement about synteny in the following sentence.

RESPONSE #1.6: Thanks for the suggestion. We have emphasized this. Please see Lines 91-94.

Lines 143-162: If space is available, a much longer Discussion section would be beneficial. Even if the journal demands brevity, the authors might still sneak in another sentence or two about the diversity-generation capacity of this type of R gene locus, possibly citing maize Rp1 or other published examples of high-diversification loci, and/or a review that does a strong job on that topic. It would be good to comment that NLR structure/function studies have recently advanced dramatically (see Science/Nature 2021 papers), and note that analysis of (extended-LRR) R6 protein multimer structure is of future interest. It would be nice to further emphasize for readers the elevated potential of highly recombinogenic R loci to generate immune diversity, and the possible ways that technologists might harness that. But it also bears note that Rps11 was not previously landed upon by other soybean geneticists (for *P. sojae* or other diseases)... A much longer Discussion is merited.

RESPONSE #1.7: We are so grateful for these comments. We have expanded the Discussion section and cited relevant work including the *Rp1* region of maize, *Pc* region of sorghum, recent *Arabidopsis* pan-NLRome, and genome sequences of a diverse maize NAM parental lines. Currently, all software including AlphaFold does not allow structural prediction of the Rps11 protein, but it is of future interest. We also mentioned potential engineering of NLR genes for resistance. Please see the highlighted text in the Discussion section (Please see Lines 216-231, 242-249).

Minor Points:

The primary literature reference choices are appropriate, but the excellent reviews/chapters that are cited are all a bit dated - is there at least one more recent source that can be used?

RESPONSE #1.8: We have now referenced additional, more recent literatures references including #10, 11, 13, 22, 23.

Please reference the literature more and/or speculate a bit about the functional purpose of the unusually long 13 kb 5'UTR.

RESPONSE #1.9: We have cited literature about 5' UTR intron functions. Please see Lines 242-245, and newly added references #28, 29

Methods should describe Fig. 4 box 4 presence absence was determined (how was a TSR defined? what allowed classification as TSR-?)

RESPONSE #1.10: Thanks for the suggestion. We have added a paragraph in the Methods for 5'RACE and TSR analysis that was missed in original submission. Please see Lines 123-124, and 395-406.

Reviewer #2 (Remarks to the Author):

Summary Review: This research group has used the traditional map-based cloning procedure to identify a single NLR gene within a cluster of NLR genes that provides resistance to a large set of *Phytophthora sojae* isolates. While the cluster was shown to provide resistance to 127 isolates, within that cluster, one gene, R6, was shown to provide resistance to three isolates by analyses of transgenic plants with the R6 gene. This appears to be a breakthrough of importance to soybean production because *P. sojae* is a major plant pathogen on the crop.

The group then goes on to discuss the evolution of the cluster where a single (or two) genes underwent tandem duplication to evolve the cluster found in the resistant genotype. This cluster was further studied in the assembled genomes of 28 other genotypes where it was found the cluster shows wide variation in NLR number and structure. I would ask, have the evolution of other clusters been described to the detail found here. If not, that should be stated. If so, a discussion of how this cluster and other clusters evolved should be included in the narrative.

RESPONSE #2.1: We would like to thank the reviewer for taking the time to assess our manuscript and for providing this set of thoughtful comments. To the best of our knowledge, none of the NLR clusters has been analyzed and compared with gap-free sequences at population level to understand rapid and dynamic NLR gene evolution and evolutionary mechanisms, but we have also discussed previous studies in a few R loci including *Rp1* in maize and *Pc* in sorghum, as well as observations from an Arabidopsis NLRome and genomic sequences from 26 diverse maize genomes. Please see Lines 216-230 and newly added references #19-23.

Research Approach: All of the experimental protocols were appropriate to draw the conclusions

reported in the manuscript. The procedures are described in enough detail for others to understand the conclusions, repeat the work, and use the outcome in applied crop improvement.

General: The authors should consider using race or isolate not both somewhat interchangeably.

RESPONSE #2.2: Thanks. For the races that have been previously defined and routinely used in other publications, we would like to use races for consistency; for those newly collected ones without race-haplotype analysis, we would use isolates.

Line 36: “subsequent Irish diaspora”; diaspora by itself generally means the movement of Jewish peoples

RESPONSE #2.3: “Irish” has been added.

Line 41: “soybean genomic regions”

RESPONSE #2.4: “soybean” has been added.

Line 65: What information suggest Rps11 “had not been fully assembled” in Williams 82?

RESPONSE #2.5: There are 12 sequencing gaps within the primary mapping interval in the Williams 82 assembly V2.0, as shown in Supplementary Fig. 3. Please also see Lines 55-57 and 72.

Line 72: What is a Rps11/rps11 region. It would be better to say “Rps11 and rps11 regions”.

RESPONSE #2.6: Thanks for the suggestion. Suggested changes have been made .

Line 75: please discuss why these NLRs are considered “large”; nucleotide length compared to what? Size of proteins compared to what?

RESPONSE #2.7: This is based on the size of the coding sequences of all NLR in the soybean reference genome. We have now added a Supplementary Fig. 10 and discussed this in Lines 171-174.

Line 117: What phylogenetic approach was used? Why was only the NB-ARC region used? This is not presented in the M&M or Fig. 2 legend.

RESPONSE #2.8: Sorry for missing this information in our original submission, which has been added in the Methods section (Please see Lines 344-350). Sequences were aligned by MEGA7

(reference #39), and the phylogenetic trees were constructed using Minimum Evolution methods (see reference #40). Only NB-ARC region was used because the NB-ARC domain is relatively conserved among all NLR genes, whereas the LRR regions are highly variable in both sequence and copy number and thus difficult to align and compare.

Line 119: "...were derived..." seems incorrect. The authors provide evidence that a single (or two) genes underwent tandem duplications to evolve the cluster. Importantly this region is distinct from the NLR cluster in the susceptible region. The team should reconsider this issue during revision.

RESPONSE #2.9: We agree with the reviewer and think that overlapped events could have occurred to make some specific events detected in one genome (e.g., PI 594527) but undetectable in another e.g., Williams). Nevertheless, we have added "likely" between "were" and "derived".

Line 128: susceptible accessions?? If so, mention here.

RESPONSE #2.10: These accessions are representative varieties either sequenced as reference genomes or selected to construct the soybean pan-genome. Most of these accessions are deposited in China seed bank and unavailable to us, and their resistance spectra are unknown.

Line 139: What does *Rps11*/*rps11* mean?

RESPONSE #2.11: We meant *Rps11* and "*rps11*" refers to the *Rps11* corresponding regions in other accessions but lacking a recessive allelic copy of *Rps11* (R6). We have changed the phrase into "*Rps11* region and "*rps11*" regions (which lacks an allelic non-functional copy of *Rps11*)" for clarity. (Please see Lines 204-205).

Line 212: What assembly (V2.0 or 3.0) of Williams 82 is being displayed.

RESPONSE #2.12: We have now mentioned Assembly V3.0 in the Line 85 and the legend of Fig. 1a.

Line 222: Relative expression of what?

RESPONSE #2.13: We did not see "Relative" in original line 222, and guess the reviewer referred it as shown in the y-axis of the plot in Fig 1d. "Relative expression" is the expression level of the examined gene (R6) in a non-root sample "relative" to that in the root sample, which was set as 1.0. (please see modified legend for Fig. 1d).

Line 255: What does D1 – D4 mean?

RESPONSE #2.14 D1-D4 refer to different duplication events detected in the region. Green arrows indicate the duplication (D1) event of R1-R2, R3-R4 and R11-R12; Orange arrows indicates the duplication (D2) event of R7, R8 and R10; Grey arrow indicates the duplication (D3) event between R9 and R11; Purple arrow indicates the duplication (D4) event between R3 and R6. We have added these description in to the legend for Fig. 3d.

Line 279: What does “one side heterozygous and the other side homozygous Williams genotype” mean. I have not heard that terminology.

RESPONSE #2.15 Two boundary markers, SSR286 and SSR320, were used to screen the mapping population. A recombinant was defined as an individual with heterozygous genotype at one boundary marker and homozygous genotype (Williams genotype or PI 594527 genotype) at the other boundary marker. For fine mapping, only recombinants with heterozygous genotype at one boundary marker and homozygous Williams genotype at the other boundary marker were used because the two expected phenotypes could be accurately distinguished (expected survival rate 0% vs 75%). We have modified the description to make it clearer (Please see Lines 257-259).

Lines 414 and 416: Who is the third party that should be contacted? Researchers need to know that to confirm the work here.

RESPONSE #2.16: This is a general statement for all publications involving Corteva germplasm. Detailed information for a specific germplasm will be provided by Corteva when materials are requested.

Reviewer #3 (Remarks to the Author):

The manuscript by Weidong Wang et al. isolated a resistant gene Rps11 in soybean by a gap-free sequence-based fine mapping and cloning approach. Rps11 is a 27.7-kb nucleotide-binding site-leucine-rich repeat gene which confers broad-spectrum resistance to *Phytophthora sojae* isolates. And Rps11 is located in a genomic region harboring a cluster of unusually large NLR genes. A comprehensive set of evolution data show that the NLR gene cluster exhibits drastic structural diversification caused by duplications and recombination. Overall, the data appear to be analyzed appropriately with one or two exceptions detailed below.

RESPONSE #3.1: We are very thankful to the reviewer for taking the time to assess our manuscript and for providing the set of valuable comments, which have helped us to improve the clarity, solidity, and readability of the manuscript.

1. This conclusion that R6 in PI 594527 is Rps11 is not justified. R6-transgene showed resistance to all tested three races. Constitutive promoter (ubiquitin promoter) was used to express R6, an

NLR gene, in an elite soybean variety 93Y21, which could have led to artificial resistance to *P. sojae* races.

RESPONSE #3.2: Thanks for the comments. We have now examined the expression levels of the transgenic *Rps11* under the regulation of the *AtUbi3* promoter and found that the expression levels of the transgenic *Rps11* were much lower than that of the native *Rps11* in a recombinant inbred line (RIL) derived from a cross between PI 594527 and Williams (please see newly added expression data in Fig. 2c; Lines 154-163). So, we believe the transgenic *Rps11* were not “overexpressed” to a level at which autoimmunity can be established. We have also examined the expression patterns of all the four NLRs in the defined *Rps11* region, and found only R6 was expressed before and after inoculation with the pathogen, and its expression was responsive to the pathogen inoculation. By contrast, the other three NLRs were not expressed either before or after inoculation (please see newly added Supplementary Fig. 6; Lines 109-120). Therefore, R6 has been justified to be *Rps11* even the transformation experiment is excluded. Additional evidence provided in our original submission includes:

- i) Fine mapping and expression analysis of parental lines and key recombinants have demonstrated that, of the four R genes (i.e., R5, R6, R7, and R8) defined in the fine-mapped region, only R6 is expressed;
- ii) Only the recombinants with R6 expression are resistant, all other recombinants without R6 expression are all susceptible.
- iii) Although the ubiquitin promoter drives constitutive expression of R6 in transgenic lines, the native R6 is also highly expressed in all the tissues examined (Fig. 1d), so the resistance observed in the transgenic lines reflect the R6 function.

Together, we believe, and sincerely hope the reviewer now agrees, our conclusion that R6 is *Rps11* is sufficiently justified. We agree with the reviewer that ideally transformation of the *Rps11* genomic DNA with its own promoter would best reveal the resistance conferred by *Rps11*. However, given such a large size of *Rps11* from the TSR to the TTS (27.7kb), the unclear boundary of its promoter region that would be somewhere upstream of the TSR, as well as anticipated difficulty in transforming such a large segment in soybean – one of the most difficult crops to be transformed even for a DNA fragment as small as few kb, we fully believe struggling for transforming a ~30-kb *Rps11* cassette with its native promoter for additional years is neither necessary/feasible nor significantly support our argument. Of course, our lines of evidence and justifications would not eliminate the limitation of our approach with the *AtUbi3* promoter, which has also been discussed (Lines 154-157, 242-245).

2. The novelty of this study was mentioned in the abstract "Our study thus exemplifies innovative evolution of NLR genes and NLR gene clusters and will accelerate the deployment of *Rps11* for soybean protection." but no further discussion of this point are mentioned in discussion section. Actually, the similar evolution of NLR genes and NLR gene clusters have been mentioned in other plants.

RESPONSE #3.3: Thanks for these good suggestions. We have now extended our discussion by including relevant studies in other plants including maize, Arabidopsis, and sorghum. Please see Lines 216-230.

3. The title is a bit exaggerated, such as "giant", "chimeric", "extreme", "a plant pathogen". It should be expressed in more precise terms.

RESPONSE #3.4: We have revised the title to make it more precise and factual.

4. Statistical significance should be presented in Fig. 1d, Fig. 2b, Fig. 2c, Extended Data Fig. 3b and Extended Data Fig. 7a.

RESPONSE #3.5: Thanks for the suggestion. We have now added the statistical significance in Fig. 2b, Fig. 2c and original Extended Data Fig. 7a (now Supplementary Fig. 8a), as well as the data points. We also added data points in newly added supplementary Fig. 6. Fig. 1d is the expression profiling of the R6 in different tissues and Extended Data Fig. 3b (now Supplementary Fig. 4b) shows which NLR genes among R1-R12 are expressed NLR, which simply if a gene is expressed or not, and thus comparisons and statistics among samples in these panels are not needed.

5. More detailed information should be provided in this manuscript, such as Rps11 sequence, isolates collected from Indiana and so on.

RESPONSE #3.6: We have added the CDS and protein sequence of *Rps11* in a new supplemental table (Supplemental Table 3) and information about the collection of isolates was added in supplemental Table 2. We have also added the geographic distribution of the collection of isolates in Supplementary Fig. 2.

6. R6 expression of a F5 RIL should be presented in Fig. 2c as a control.

RESPONSE #3.7: Thanks. This is a great suggestion and we have added the R6 expression of the F5 RIL in modified Fig. 2c.

Reviewers' Comments:

Reviewer #1:

Remarks to the Author:

I am satisfied that the authors have acceptably addressed all of the comments of all three reviewers. In particular, regarding reviewer 3 point 3.1, the authors have done an excellent job of meeting more than enough criteria to conclude that R6 is the correct gene. The manuscript remains a strong contribution.

My remaining concerns all surround the fact that that the correction edits seem to have been done without benefit of a native English speaker (there are many on the author team). Many sentences in the added/revised text are a bit rough with regard to English syntax, but the meanings can be discerned. Lines 154-163, and 242-245, are particularly in need of editing.

line 115 Excessive confidence. Should say "Therefore, R6 is the only "likely" candidate for Rps11" (add "likely")

line 119 Excessive confidence. Should say "Together, these observations "strongly suggest" that R6 is indeed Rps11.

Reviewer #2:

Remarks to the Author:

I have reviewed the response to my original questions and found them satisfactory. The answers have improved the manuscript.

Reviewer #3:

None

REVIEWERS' COMMENTS

Reviewer #1 (Remarks to the Author):

I am satisfied that the authors have acceptably addressed all of the comments of all three reviewers. In particular, regarding reviewer 3 point 3.1, the authors have done an excellent job of meeting more than enough criteria to conclude that R6 is the correct gene. The manuscript remains a strong contribution.

My remaining concerns all surround the fact that that the correction edits seem to have been done without benefit of a native English speaker (there are many on the author team). Many sentences in the added/revised text are a bit rough with regard to English syntax, but the meanings can be discerned. Lines 154-163, and 242-245, are particularly in need of editing.

RESPONSE: Thanks for pointing this out. Our coauthors who are native English speakers polished the language.

line 115 Excessive confidence. Should say "Therefore, R6 is the only "likely" candidate for Rps11" (add "likely")

RESPONSE: We have added "likely" in this sentence.

line 119 Excessive confidence. Should say "Together, these observations "strongly suggest" that R6 is indeed Rps11.

RESPONSE: We have revised the sentence as suggested.

Reviewer #2 (Remarks to the Author):

I have reviewed the response to my original questions and found them satisfactory. The answers have improved the manuscript.

[Editor: **Reviewer #3** states that (s)he is satisfied with the revision.